# Production of 2,3-Butanediol by *S. cerevisiae* L7 in Fed-Batch Fermentation with Optimized Culture Conditions

**Guoxu Ao** [1,†], **Shanshan Sun** [1,†], **Lei Liu** [1], **Yuhao Guo** [1], **Xiujun Tu** [1], **Jingping Ge** [1,2,*] and **Wenxiang Ping** [1,2,*]

[1] Engineering Research Center of Agricultural Microbiology Technology, Ministry of Education & Heilongjiang Provincial Key Laboratory of Plant Genetic Engineering and Biological Fermentation Engineering for Cold Region & Key Laboratory of Molecular Biology, College of Heilongjiang Province & School of Life Sciences, Heilongjiang University, Harbin 150080, China
[2] Hebei Key Laboratory of Agroecological Safety, Hebei University of Environmental Engineering, Qinhuangdao 066102, China
[*] Correspondence: gejingping@126.com (J.G.); wenxiangp@aliyun.com (W.P.); Tel.: +86-0451-86609106 (J.G.); Fax: +86-0451-86608046 (J.G.)
[†] These authors contributed equally to this work and share the first authorship.

**Abstract:** As a chemical platform, 2,3-Butanediol (2,3-BD) has been widely applied in various industrial fields. In this study, to enhance the production of 2,3-BD by *Saccharomyces cerevisiae* L7, Plackett–Burman (PB) multifactorial design, the steepest climb test and central composite design (CCD) were employed to optimize the culturing conditions of *S. cerevisiae* L7. The results show that acetic acid, peptone and glucose were contributing factors for 2,3-BD production. Subsequently, a satisfactory production of 2,3-BD (13.52 ± 0.12 g/L) was reached under optimal conditions, which was 3.12 times higher than before optimization. Furthermore, fed-batch fermentation was carried out under optimized culture conditions, and a higher production and yield efficiency of 2,3-BD were achieved (21.83 ± 0.56 g/L and 0.15 ± 0.01 g/g, respectively) when glucose (20 g/L) and acetic acid (0.2 g/L) were added at 12, 24, 36, 48 and 60 h. Therefore, the production and yield efficiency of 2,3-BD were higher than those without fed-batch fermentation (61.46% and 58.51%, respectively). These results provide good support and a technical foundation for the large-scale industrial production of 2,3-BD by *Saccharomyces cerevisiae*.

**Keywords:** 2,3-BD; *Saccharomyces cerevisiae*; Plackett–Burman design; central composite design; fed-batch fermentation

## 1. Introduction

2,3-Butanediol (2,3-BD) is an important chemical raw material and liquid fuel that is listed as a platform chemical with great potential for industrial application and is widely used in food, agriculture, medicine, aerospace and other fields [1–4]. Compared with physical and chemical techniques, the microbiological production of 2,3-BD has several advantages, including relatively high efficiency and ecofriendly characteristics [5–7]. Naturally, many microorganisms can produce 2,3-BD, such as *Klebisella* sp., *Serratia* sp., *Pseudomonas* sp., *Aerobacter* sp. and *Saccharomyces* sp. [8–11]. Among them, most microorganisms produce lactic acid, acetic acid and other byproducts, resulting in an increased cost of product recovery and purification [12].

As a generally regarded as safe (GRAS) microorganism, *S. cerevisiae* is widely used in the production of various biological chemicals and biofuels and has a strong tolerance to ethanol and harsh conditions [5]. It is also considered important in the potential production of 2,3-BD [13,14]. However, wild-type *S. cerevisiae* W141 achieved low production of 2,3-BD, with only 1.95 g/L [15]. Changing the culture conditions of 2,3-BD producers can overcome the disadvantage of low yield of the original strain [16]. When the fermentation temperature, pH and inoculum size were optimized for *Klebsiella pneumoniae* at 37 °C,

pH 6.1 and 1.0 (%$w/v$), respectively, 2,3-BD production increased by 44% [17]. Optimizing 2,3-BD production by *S. cerevisiae*, the 2,3-BD production reached 3.25 ± 0.03 g/L with 1.0 g/L acetic acid and 4.52 mg/L dissolved oxygen [13].

Response surface methodology (RSM), which can reveal the interactions and correlation between multiple components of the medium and environmental factors, has been widely used to improve the production of 2,3-BD [18]. The use of RSM can not only screen the composition of the medium but also improve the culture conditions. When the pH, aeration and corn pulp concentration were optimized to 5.6, 3.50 vvm and 45.0 mL/L, respectively, the production of 2,3-BD by *Klebsiella pneumoniae* Δ*ldhA* could be increased to 2.48 g/L [19]. In the case of *Enterobacter ludwigii*, the levels of yeast extract, KOH and $(NH_4)_2SO_4$ were optimized to 10 g/L, 6.2 g/L and 0.7 g/L, respectively, and the 2,3-BD yield increased to 0.31 g/g [20]. Therefore, optimizing the culture conditions and selecting the appropriate fermentation strategy for 2,3-BD producers has good potential for improving the production of 2,3-BD.

Remarkably, during the initial fermentation process, a higher glucose concentration exerted greater osmotic pressure on the cells, resulting in the separation of the cytoplasmic wall and affecting glucose transport within the cells. Moreover, according to the Crabtree effect [21–23], a high glucose concentration may inhibit the respiration of *S. cerevisiae*, thus reducing the cell production rate [24,25]. In a previous study, when the glucose concentration exceeded 5%, the synthesis of respiratory enzymes and the formation of mitochondria were inhibited in cells. *S. cerevisiae* has a strong Crabtree effect during the fermentation process, which is affected by both dissolved oxygen and sugar concentration during the sugar fermentation process [13,26,27]. Hence, to obtain a higher output of target products, the initial sugar concentration needs to be strictly controlled to obtain 2,3-BD production. Wild-type *S. cerevisiae* L7, which has the capacity for 2,3-BD production, was obtained from black soil in Heilongjiang Province, China, and was preserved at the Key Laboratory of Microbiology, Heilongjiang University, China.

To enhance the original low production (4.33 g/L) of 2,3-BD by *S. cerevisiae* L7, the strategy of changing the medium composition and optimizing the culture conditions is worth considering. In this study, appropriate amounts of acetoin and acetic acid were added to the culture medium at the beginning of fermentation by *S. cerevisiae* L7, and a single-factor test was performed to optimize the carbon source, nitrogen source, shaking speed, type of corks, acetic acid, $(NH_4)_2SO_4$ and $Na_2EDTA$. Furthermore, the Plackett–Burman (PB) design and central composite design (CCD) were employed to obtain the optimal fermentation conditions for the production of 2,3-BD [28–30]. It was hypothesized that reasonable design and combinations, such as optimal carbon sources and cultural conditions, could increase the activity of key enzymes in metabolic pathways and enhance the yield of the end products. The purpose of this study was to (1) screen the factors that have a great influence on the production of 2,3-BD, (2) establish a 2,3-BD fermentation model using RSM and (3) explore the effect of fed-batch culture technology on the production of 2,3-BD. These results will provide experimental support and technical assurance for future large-scale industrial production.

## 2. Materials and Methods

### 2.1. Strain and Cultivation

*S. cerevisiae* L7 was isolated from black soils of Heilongjiang Province, China, and preserved in the Key Laboratory of Microbiology at Heilongjiang University as a diploid wild-type strain. The seed of *S. cerevisiae* L7 was cultured in YPD medium, which was composed of (g/L) yeast extract, 10; peptone, 10; and glucose, 20. The solution was cultured on a rotary shaker at 140 rpm for 12 h at 37 °C. The composition of the fermentation medium was as follows (g/L): glucose, 80; peptone, 20; yeast extract, 10; YNB, 3.4; $K_2HPO_4$, 3; $KH_2PO_4$, 12; $(NH_4)_2SO_4$, 10; and $Na_2EDTA$, 1.5. Acetic acid (1 g/L) was added at the beginning of fermentation. The cultivation was performed at 30 °C with a shaking speed of 140 rpm in 250 mL flasks with a working volume of 100 mL for 72 h.

### 2.2. Single-Factor Experiment

The carbon source concentration, nitrogen source concentration, $Na_2EDTA$ concentration, acetic acid concentration and dissolved oxygen content were studied by using the "one factor at a time" method (keeping the remaining factors unchanged and maintained at the optimal level), and the production of 2,3-BD was used as the detection index to screen out the optimal fermentation conditions of the strain (in this experiment, a two-stage speed control was adopted to change the oxygen environment by switching between high (140 rpm at 0–18 h) and low speeds (80 rpm at 18–120 h) to accumulate more 2,3-BD).

### 2.3. Plackett–Burman Design

In this study, RSM was applied to improve 2,3-BD production based on single-factor experimental data (detailed information was shown in the Supplementary Material). Five factors were selected, including glucose, peptone, $(NH_4)_2SO_4$, acetic acid and shaking speed. The independent variables were converted into coded values for computational convenience: the upper limit of a factor was coded as +1, the lower limit as −1 and the center level as 0 (Table 1). Fifteen runs were carried out according to the experimental layout (Table S1) and 2,3-BD production was used as the response. All the runs were performed in triplicate. Design Expert (version 7.0.0, Stat-Ease, Minneapolis, MN, USA) was employed for statistical analysis of the PB design. Regression analysis determined the factors that had a significant ($p < 0.01$) effect on 2,3-BD production, and these factors were evaluated in further optimization experiments.

**Table 1.** The factorial levels and the regression analysis of the Plackett–Burman design.

| No. | Factors | Coded Level | Central Point | Coded Level | Coefficient | Effect | T-Value | *p*-Value |
|-----|---------|-------------|---------------|-------------|-------------|--------|---------|-----------|
| | | +1 | 0 | −1 | | | | |
| $X_1$ | Glucose (g/L) | 130 | 120 | 110 | 0.2517 | 0.503 | 2.21 | 0.069 |
| $X_2$ | Peptone (g/L) | 17 | 15 | 13 | 0.3533 | 0.707 | 3.11 | 0.021 * |
| $X_3$ | $(NH_4)_2SO_4$ (g/L) | 7 | 6 | 5 | 0.0417 | 0.083 | 0.37 | 0.726 |
| $X_4$ | Acetic acid (g/L) | 1.2 | 1.0 | 0.8 | −0.6183 | −1.237 | −5.44 | 0.002 ** |
| $X_5$ | Shaker speed (rpm) | 90 | 80 | 70 | 0.0367 | 0.073 | 0.32 | 0.758 |

$R^2$ = 0.8810; adjusted $R^2$ = 0.7819; adequate precision 9.0900 * $p < 0.05$ indicated significant differences; ** $p < 0.01$ indicated extremely significant differences.

### 2.4. The Steepest Climb Test Design

The steepest climb test was applied to narrow the range of significant factors selected by the PB design. The values of each variable in the steepest climb test were the central point of the PB test, and the climb direction was the positive and negative effects of each significant factor. The isogradient increasing factor values were taken when the significant factor was positive; conversely, when the significant factors had a negative effect, the isogradient reducing factor values were taken. The climbing test was carried out until the peak production of 2,3-BD was reached, which was the central point of the subsequent central composite design (CCD) experiment.

### 2.5. Central Composite Design

The results of the PB screened three factors that had a significant effect on the production of 2,3-BD by *S. cerevisiae* L7, which were $X_1$ (glucose), $X_2$ (peptone) and $X_4$ (acetic acid). Based on the effect values of each factor, a CCD experiment was conducted using Design Expert 11 software, wherein five levels (−1.682, −1, 0, +1 and +1.682) were selected for each factor and the factor levels were set as shown in Table 1. The factors studied in the CCD and the actual levels and results of the CCD are shown in Table S2. Further analysis

of the experimental data for second-order polynomials assuming quadratic, linear and interacting effects is shown in Equation (1).

$$Y = b_0 + \sum_{i=1}^{n} b_i X_i + \sum_{j=1}^{n} b_{jj} X_j^2 + \sum_{i<j}^{n} b_{ij} X_i X_j \tag{1}$$

$Y$ is the response value, $X$ is the input variable, $b_0$ is the intercept coefficient, $b_i$ is the linear coefficient, $b_{jj}$ is the quadratic coefficient and $b_{ij}$ is the interaction coefficient.

### 2.6. Fed-Batch Fermentation

To avoid the Crabtree effect caused by high concentrations of glucose and improve the regulatory effect of acetic acid on *S. cerevisiae* L7, fed-batch fermentation was performed to maintain 2,3-BD production by intermittent feeding at the most appropriate time under optimal culture conditions. To optimize the feeding time, 0.36 g/L acetic acid and 20 g/L glucose were simultaneously added at 12 h, 16 h, 20 h and 24 h. Then, 0.1, 0.2, 0.3, 0.4 and 0.5 g/L acetic acid were added at 12 h to determine the optimal acetic acid concentration. The cultivation was performed at 30 °C with a shaking speed of 140 rpm in 250 mL flasks with a working volume of 100 mL, and samples were taken every 12 h. Dry cell weight (DCW), pH value and production of 2,3-BD were used as indices to determine the appropriate time of acetic acid addition.

### 2.7. Analytical Methods

The concentrations of glucose, 2,3-BD, acetic acid and ethanol were determined using a high-performance liquid chromatography (HPLC) system (10 AVP HPLC, Shimadzu Co., Kyoto, Japan) with an Aminex HPX-87H column (300 mm × 7.8 mm) (Bio-Rad, Palo Alto, CA, USA) at 65 °C equipped with an RID10A refractive index detector. The mobile phase was 0.005 mol/L $H_2SO_4$ solution at 0.6 mL/min.

### 2.8. Statistical Analysis

The data in this study were all obtained from three independent experiments, and the mean value ± standard deviation (SD) was presented. The experimental design was performed with the statistical software Design Expert (version 7.0.0, Stat-Ease, Minneapolis, MN, USA). In addition, JMP 9.0.2 (SAS Institute Inc., Cary, NC, USA) software was used to analyze statistical data.

## 3. Results and Discussion

### 3.1. Effects of Single Factors on the Production of 2,3-BD

Glucose is a widely existing carbon source in nature, and the demand for cyclic oxidative phosphorylation of the tricarboxylic acid (TCA) cycle in cells is negatively correlated with the concentration of glucose [31,32]. As shown in Figure 1A,B, glucose and peptone both had a significant effect on the production of 2,3-BD by *S. cerevisiae* L7 ($p < 0.0001$), and the production of 2,3-BD first increased and then decreased with increasing concentrations of glucose and peptone, respectively. When the concentrations of glucose and peptone were 120 g/L and 15 g/L, the production of 2,3-BD reached 4.51 ± 0.08 g/L and 5.11 ± 0.07 g/L, respectively. As the effect of $(NH_4)_2SO_4$ on the production of 2,3-BD by *S. cerevisiae* L7 was not significant (Figure 1C, $p > 0.05$), the concentration of $(NH_4)_2SO_4$ was selected as 6 g/L to maintain the appropriate C/N ratio of the medium and save consumption.

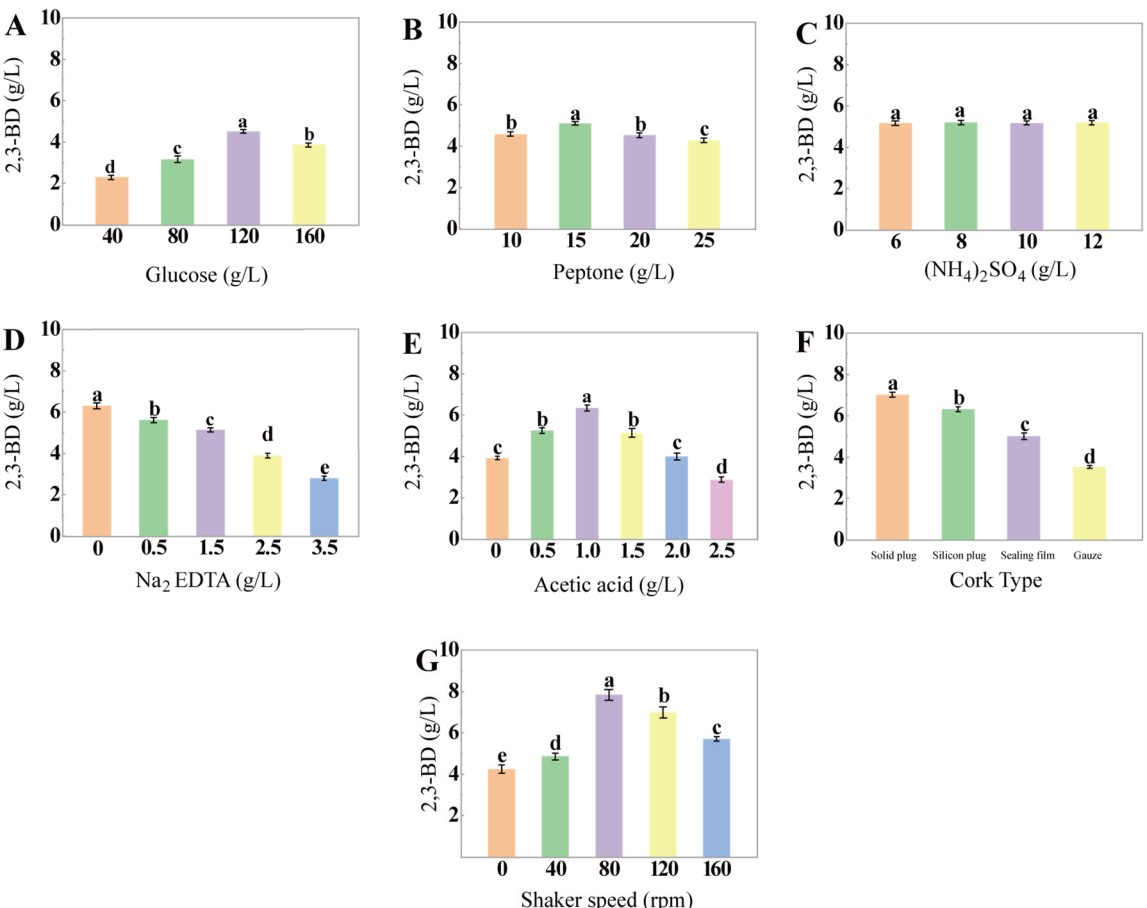

**Figure 1.** Effects of a single factor on 2,3-BD production of *Saccharomyces cerevisiae* L7. (**A**) Glucose; (**B**) Peptone; (**C**) $(NH_4)_2SO_4$; (**D**) $Na_2EDTA$; (**E**) Acetic acid; (**F**) Cork type; (**G**) Shaker speed (different letters indicate significant differences).

$Na_2EDTA$ showed a significantly negative effect on the production of 2,3-BD by *S. cerevisiae* L7 ($p < 0.0001$); however, $Na_2EDTA$ could promote *K. pneumoniae* production of 2,3-BD at 1.2 g/L [33]. These results indicated that $Na_2EDTA$ had different effects on 2,3-BD production by different microorganisms; therefore, $Na_2EDTA$ was not subsequently considered. According to Figure 1E, the concentration of acetic acid significantly affected the production of 2,3-BD ($p < 0.0001$), and the largest production was achieved ($6.34 \pm 0.14$ g/L) with the addition of 1 g/L acetic acid. Different cork types, which represented different concentrations of dissolved oxygen, had a significant impact on the production of 2,3-BD ($p < 0.0001$), and because the production of 2,3-BD was negatively correlated with the cork type, solid corks were selected for this experiment (Figure 1F).

Figure 1G displayed the effect of the flask shaking speed on the production of 2,3-BD, which was 11.68% higher at the two-stage shaking speed than at the constant shaking speed (detailed information is shown in Figure S1), which proved that the two-stage shaking speed could be applied to the production of 2,3-BD by *S. cerevisiae* L7 [34,35], and the production of 2,3-BD was the highest when the shaking speed was 80 rpm in the later stage. Depending on the results of single-factor experiments, the initial medium composition of *S. cerevisiae* L7 to produce 2,3-BD was 120 g/L glucose, 15 g/L peptone, 6 g/L $(NH_4)_2SO_4$ and 1 g/L acetic acid with a solid stopper to regulate the optimal oxygen content.

### 3.2. Factors with Significant Effects on 2,3-BD Production of S. cerevisiae L7

Five factors, including glucose, peptone, acetic acid, cork type and shaking speed, were used to select the significant effects on 2,3-BD production in the PB experiment. The experimental design and the values of 2,3-BD production for the fifteen runs were

presented in Tables 1 and S1, as well as demonstrating the regression analysis and *p*-values of the experimental factors. The 2,3-BD production of *S. cerevisiae* L7 ranged from $6.73 \pm 0.09$ to $9.14 \pm 0.19$ g/L under different conditions, which illuminated a significant effect of the selected factors on 2,3-BD production. Moreover, the $R^2$ (0.8810) and the adjusted $R^2$ (0.7819) clearly indicated that the observed values were in good agreement with the model predictions. In detail, the *p*-value of acetic acid (0.002) was less than 0.01, suggesting that it had extremely significant effects on 2,3-BD production. The *p*-values of peptone (0.021) and glucose (0.069) were between 0.01 and 0.05, implying that they had a significant effect on 2,3-BD production. The coefficient of acetic acid was negative (−0.6183, Table 1), which meant that acetic acid was negatively related to 2,3-BD production. Nitrogen was the second most important nutrition source just prior to the carbon source for microbial growth and metabolism. In the single-factor experiment, $(NH_4)_2SO_4$ displayed no significant effect on 2,3-BD production; however, peptone showed significant effects on 2,3-BD production. These reports collectively suggest that whether and how nitrogen sources exerted impacts on 2,3-BD production depended on microbial species, culture conditions and methods. Additionally, combined with the Pareto chart and regression coefficients (Figures S2 and S3), glucose ($X_1$), peptone ($X_2$) and acetic acid ($X_4$) were identified as the most influential factors on 2,3-BD production.

The steepest climbing experiment was essential to assess the optimal level of the chosen factors after the screening procedure with PB based on statistical analysis. In this study, the *p*-value (0.1869) in the test of equality variances (Table S3) was more than 0.05, which indicated that, in the *t*-test (Table S4), equal variance results should be chosen to assess the necessity of the steepest ascent experiment. The *p*-value (0.2997) of the *t*-test implied that the data between the central point (trial Nos. 13, 14, and 15 in Table S1) and the experimental point (trial Nos. 1–12 in Table S1) were not significantly different, indicating that the central point in the two-level factorial design had not yet approached the maximal response region and that the steepest climbing experiment was necessary [36].

Table 2 showed the steepest climbing test design and the corresponding results for each experiment. The highest 2,3-BD production was found in Run 5 ($p < 0.05$), and this result was closer to the highest 2,3-BD production range, so the center point of this experimental CCD was chosen to be the Run 5 time [37].

**Table 2.** Experimental design and results of the steepest climbing.

| Run | Acetic Acid (g L$^{-1}$) | Glucose (g L$^{-1}$) | Peptone (g L$^{-1}$) | 2,3-BD Production (g/L) |
|---|---|---|---|---|
| Origin | 1 | 120 | 15 | $7.14 \pm 0.08$ |
| 1 | 0.95 | 125 | 16 | $7.39 \pm 0.11$ |
| 2 | 0.90 | 130 | 17 | $9.10 \pm 0.15$ |
| 3 | 0.85 | 135 | 18 | $9.28 \pm 0.13$ |
| 4 | 0.80 | 140 | 19 | $9.87 \pm 0.09$ |
| 5 | 0.75 | 145 | 20 | $11.83 \pm 0.18$ |
| 6 | 0.70 | 150 | 21 | $11.45 \pm 0.21$ |
| 7 | 0.65 | 155 | 22 | $11.17 \pm 0.15$ |
| 8 | 0.60 | 160 | 23 | $10.54 \pm 0.16$ |

### 3.3. Optimal Conditions for 2,3-BD Production

The CCD design was employed for further optimization of 2,3-BD production based on the results of the PB design and the steepest climbing experiment, and a three-factor CCD with nineteen experimental runs was carried out (Table S2). The observed 2,3-BD production varied from $8.16 \pm 0.11$ to $13.52 \pm 0.21$ g/L. ANOVA (analysis of variance) is usually used as one of the methods for model fit testing. The ANOVA of the quadratic regression model not only showed that the model was highly significant but also a very low probability value ($p < 0.0001$) of the F test (F value = 15.15). The coefficient of determination ($R^2 = 0.9381$) explained that 93.81% of the variability was attributable to the variation in the independent variables. The adjusted coefficient of determination (adjusted $R^2 = 0.8761$)

corroborated the significance of the model. Moreover, the lack of fit test ($p > 0.05$) could be used for further analysis due to the statistical significance of the quadratic model.

Regression analysis was conducted (Table 3) based on the CCD results (Table S2). The coefficients of the three factors quadratic terms were all negative ($X_1^2 = -1.01$, $X_2^2 = -1.02$ and $X_4^2 = -0.7207$), suggesting that the three-dimensional response surface was openside-down and had a maximal response value. The final equation predicting the maximal 2,3-BD production in terms of coded factors was as follows:

$$Y = 13.33 - 0.7253X_1 + 0.5091X_2 + 0.1218X_4 - 0.7200X_1X_2 - 0.5100X_1X_4 - 0.5400X_2X_4 - 1.01X_1^2 - 1.02X_2^2 \\ -0.7207X_4^2 \tag{2}$$

where $Y$ is the predicted value of 2,3-BD production, $X_1$ is the glucose content, $X_2$ is the peptone content and $X_4$ is the acetic acid content.

**Table 3.** ANOVA results for the 2,3-BD production response [a].

| Source | Coefficient | DF | SS | MS | *F*-Value | *p*-Value |
|---|---|---|---|---|---|---|
| Model | | 9 | 47.15 | 5.24 | 15.15 | 0.0002 [b] |
| $X_1$ | −0.7253 | 1 | 7.18 | 7.18 | 20.78 | 0.0014 [b] |
| $X_2$ | 0.5091 | 1 | 3.54 | 3.54 | 10.24 | 0.0108 [b] |
| $X_4$ | 0.1218 | 1 | 0.2026 | 0.2026 | 0.5860 | 0.4636 [c] |
| $X_1X_2$ | −0.7200 | 1 | 4.15 | 4.15 | 11.99 | 0.0071 [b] |
| $X_1X_4$ | −0.5100 | 1 | 2.08 | 2.08 | 6.02 | 0.0366 [b] |
| $X_2X_4$ | −0.5400 | 1 | 2.33 | 2.33 | 6.75 | 0.0289 [b] |
| $X_1^2$ | −1.01 | 1 | 13.94 | 13.94 | 40.32 | 0.0001 [b] |
| $X_2^2$ | −1.02 | 1 | 14.34 | 14.34 | 41.45 | 0.0001 [b] |
| $X_4^2$ | −0.7207 | 1 | 7.09 | 7.09 | 20.50 | 0.0014 [b] |
| Residual | | 9 | 3.11 | 0.3458 | | |
| Lack of Fit | | 5 | 2.90 | 0.5805 | 11.06 | 0.0186 |
| Pure Error | | 4 | 0.2099 | 0.0525 | | |
| Total Error | | 18 | | | | |

[a] Note: $R^2 = 0.9381$; coefficient of variation (CV) = 5.18%; adjusted $R^2 = 0.8761$. DF =degrees of freedom; SS = sum of squares; MS =mean square. $X_1$ = glucose; $X_2$ =peptone; $X_4$ =acetic acid; [b] Significant at "*p*-value" less than 0.05. [c] Insignificant at "*p*-value" more than 0.05.

The interactions between variables were usually visualized with contour plots, and the ellipse plots showed significantly stronger interactions between variables than the circles (Figure 2). 2,3-BD production was varied by subtle changes in variables, with optimized values of 145.58 g/L, 21.58 g/L and 0.72 g/L for glucose, peptone and acetic acid, respectively, predicting a 2,3-BD production of 13.63 g/L.

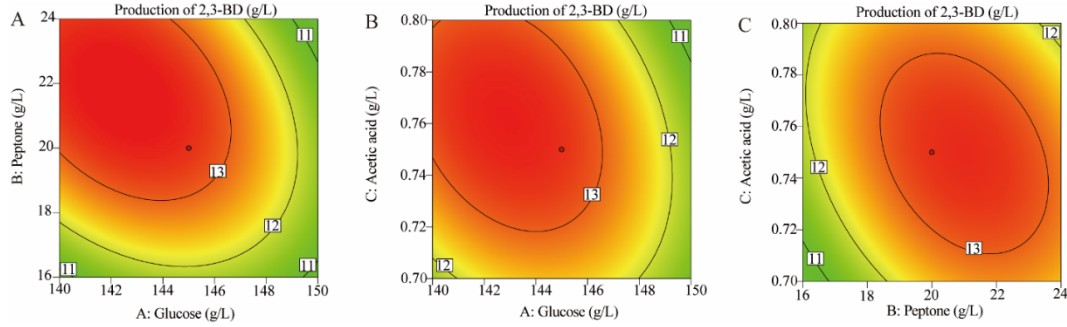

**Figure 2.** Response contour showing the effects and interactions of (**A**) glucose and peptone on *S. cerevisiae* L7 2,3-BD production, (**B**) acetic acid and glucose on *S. cerevisiae* L7 2,3-BD production and (**C**) peptone and acetic acid on *S. cerevisiae* L7 2,3-BD production.

Under optimal culturing conditions, a verification test was conducted (Table S5). The *t*-test showed that there was no significant difference ($p > 0.05$) between the predicted value of 13.63 g/L and the observed value of 13.52 ± 0.12 g/L.

### 3.4. Effects of Interactive Factors

RSM significantly revealed the interrelationships between the factors and the effect on 2,3-BD production with 2,3-BD as the response value and a two-dimensional response plot depicting the significant interactions (Table 3). As shown in Figure 2, one factor remained at the central level, while the other two factors varied over the range of the experiment. The interactive effects between glucose and peptone, glucose and acetic acid as well as peptone and acetic acid indicated that 2,3-BD production was highly affected by acetic acid compared to the other two factors.

As shown in Figure 2A, the amount of peptone resulted in changes in 2,3-BD production, and the interaction between peptone and glucose had a positive effect on 2,3-BD production. At a given concentration of glucose and acetic acid, 2,3-BD production varied with different peptone additions. The response value increased with increasing peptone, and the 2,3-BD production was 11.60 g/L when the peptone concentration was 16 g/L; however, the 2,3-BD production reached a maximum of 13.52 g/L when the peptone concentration was 20 g/L. This implied that a certain concentration of peptone was positively correlated with 2,3-BD production. The results of this experiment were similar to those of Yu et al., who showed that 2,3-BD production was 36.7 g/L after controlling the concentration of peptone in the fermentation system by *Bacillus licheniformis* [38]. This suggested that the addition of peptone could effectively increase the production of 2,3-BD.

The contours in Figure 2B show that 2,3-BD production increased to a maximum and then decreased with the addition of glucose and acetic acid. The maximum production of 2,3-BD was 13.52 g/L when 145 g/L glucose was added, indicating that the asymptotic value of 2,3-BD production had been reached with the addition of glucose. In addition, Narisetty et al. used *Escherichia coli* for 2,3-BD production by fed-batch fermentation with glucose as the substrate, resulting in a final production of 144.5 g/L [39]. These results suggested that the introduction of appropriate concentrations of glucose may improve 2,3-BD production.

As shown in Figure 2C, the interaction between acetic acid and peptone addition had a positive effect on the production of 2,3-BD. For a given peptone concentration, 2,3-BD production changed significantly with increasing acetic acid addition. For example, when the concentrations of peptone and acetic acid were 20 g/L and 0.70 g/L, the production of 2,3-BD was 12.52 g/L. When the concentration of acetic acid reached 0.78 g/L, the production of 2,3-BD was 13.40 g/L. Therefore, within a certain range, when the concentrations of peptone and glucose were constant, the production of 2,3-BD increased with increasing acetic acid concentration.

### 3.5. Fed-Batch Fermentation of 2,3-BD in a Rotary Shake-Flask

To avoid the Crabtree effect of *S. cerevisiae*, the initial glucose concentration in the fermentation medium after RSM optimization was set at 50 g/L, and 2,3-BD was detected as early as 24 h, earlier than that at 80 g/L (Figures 3A and S1B), indicating that an appropriate reduction in glucose concentration could alleviate metabolic inhibition and favor the production and accumulation of 2,3-BD. In this study, 0.36 g/L acetic acid and 20 g/L glucose were fed every 12 h when the fermentation proceeded to 12, 16, 20 and 24 h, and the residual glucose and the production of 2,3-BD were detected to observe the effects of different addition times on the production of 2,3-BD by *S. cerevisiae* L7 (samples were taken at 12 h intervals during fermentation).

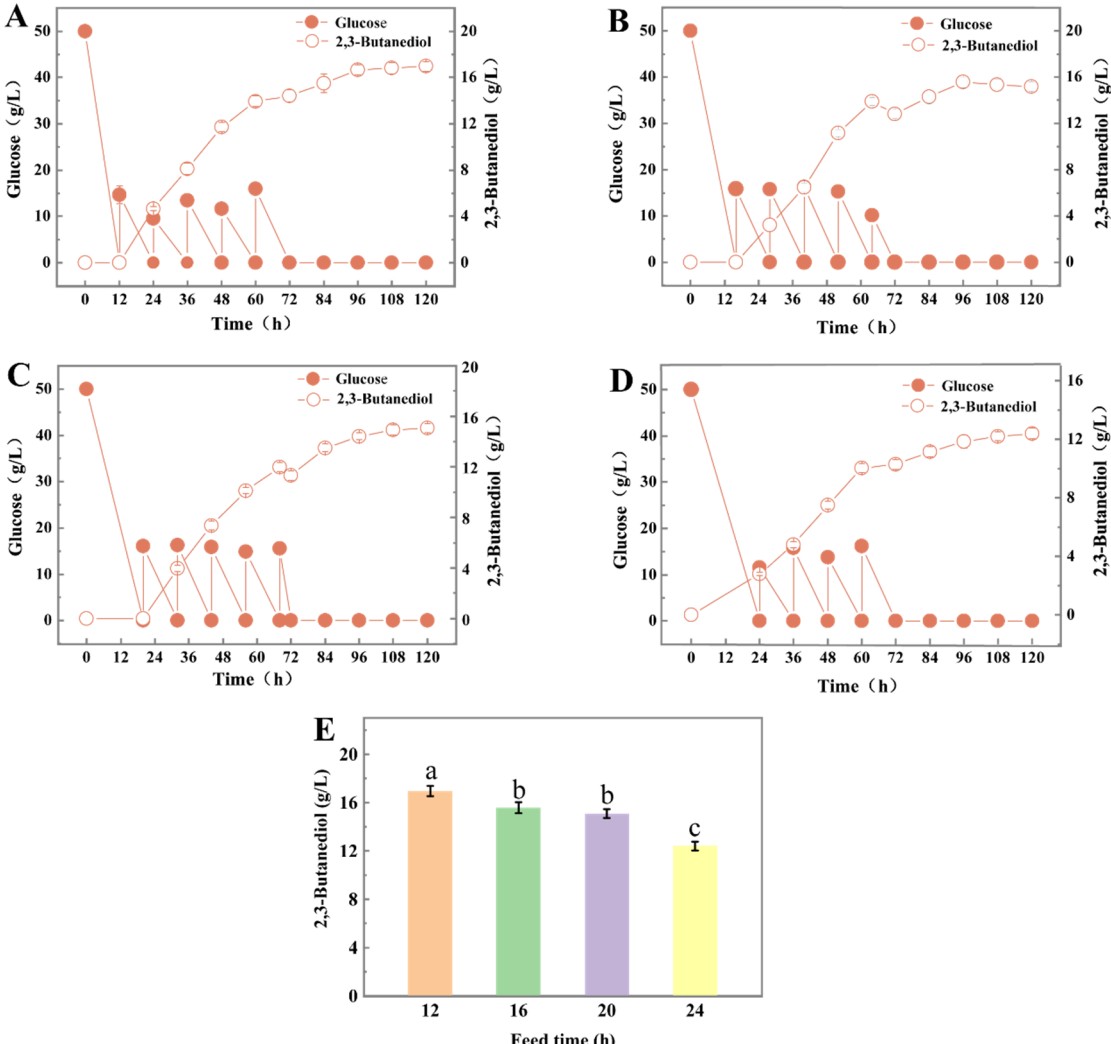

**Figure 3.** The effects of different feeding timings on the production of 2,3-BD by *S. cerevisiae* L7; (**A**–**D**) Glucose and acetic acid added simultaneously at 12, 16, 20 and 24 h, respectively; (**E**) Multiple comparisons of 2,3-BD production under different acetic acid feed timings (significant differences between different letter variables).

The trends in 2,3-BD production were roughly similar with different feeding modes, and the production of 2,3-BD in the feeding stage was greater than that without feeding (Figure 3 and Table S5). Regardless of whether glucose and acetic acid were added simultaneously at 64 or 68 h (Figure 3B,C), the concentration of 2,3-BD decreased at 72 h, which may be due to the gradual production of harmful metabolites by microorganisms as fermentation time prolonged, leading to a decrease in cellular metabolic activity and entering a decline period. At the same time, adding acetic acid to the fermentation medium exacerbated the deterioration of the yeast's living environment and was not conducive to the production of 2,3-BD. Moreover, acetic acid was added to the fermentation medium, which aggravated the deterioration of the survival environment for *S. cerevisiae* L7 and was unsuitable for the production of 2,3-BD.

Obviously, Figure 3D shows that the production of acetic acid and glucose supplementation at 24 h is lower than that of unfed 2,3-BD, indicating that the production of 2,3-BD in *S. cerevisiae* L7 in a certain time range (12 h, 16 h, 20 h and 24 h) was inversely correlated with feeding time, mainly because, with the change in time, the metabolic capacity of the strain decreases, and then supplementation of acetic acid would produce acid stress, which was not conducive to the production of 2,3-BD.

Multiple comparisons between 2,3-BD production at different times of acetic acid addition were analyzed in Figure 3E and Table 4, which showed that different addition times of acetic acid had different effects on 2,3-BD production. When glucose and acetic acid were fed at 12 h for the first time, the production and yield of 2,3-BD reached the maximum, which increased by 25.44% ($13.52 \pm 0.12$ g/L) and 20.21% ($0.094 \pm 0.003$ g/g) compared with those without feeding fermentation, respectively. Additionally, the single-cell production intensity achieved a maximum value of $2.162 \pm 0.007$ g/g/g. Therefore, 12 h was selected as the glucose and acetic acid addition time point for the following study.

**Table 4.** Production of 2, 3-butanediol under different feeding times.

| Timing of Additions | 2,3-BD Production (g/L) | Yield of 2,3-BD (g/g) | Single-Cell Production Intensity (g/g/g) |
|---|---|---|---|
| 12 h | $16.96 \pm 0.44$ | $0.113 \pm 0.003$ | $2.162 \pm 0.007$ |
| 16 h | $15.57 \pm 0.46$ | $0.103 \pm 0.003$ | $1.983 \pm 0.009$ |
| 20 h | $15.08 \pm 0.36$ | $0.101 \pm 0.002$ | $1.997 \pm 0.010$ |
| 24 h | $12.40 \pm 0.36$ | $0.095 \pm 0.003$ | $1.875 \pm 0.009$ |

Different concentrations of acetic acid (0.1, 0.2, 0.3, 0.4 and 0.5 g/L) and 20 g/L glucose were fed at 12 h during the fermentation process, which showed that different concentrations of acetic acid had a significant effect on the production of 2,3-BD ($p < 0.05$). Figure 4 showed that, with increasing concentrations of acetic acid feeding, the production of 2,3-BD by *S. cerevisiae* L7 gradually increased. Compared with 0.1 g/L acetic acid feeding (Figure 4A), 0.2 g/L acetic acid feeding increased 2,3-BD production by 29.55% ($16.85 \pm 0.57$ g/L, $21.83 \pm 0.56$ g/L) at 120 h, and the yield increased by 33.04% (Table 5 and Figure 4B). However, when the acetic acid concentration continued to increase (0.3–0.5 g/L), the production of 2,3-BD appeared to decrease compared to that at 0.2 g/L acetic acid feeding (Figure 4C–E). This result was similar to that of Liu et al. and indicated that the addition of acetic acid at an appropriate concentration could enhance 2,3-BD production [13]. Although the exogenous feeding of acetic acid could increase the production of 2,3-BD, when it was added into the culture medium with a pH below the dissociation constant of acidic substances (pKa 4.76), it could enter into the cell and dissociate into $H^+$ and acid ions. If the concentration of acetic acid feeding was exceeded, it would first cause cell acid stress and consume a large amount of intracellular ATP, which in turn inhibited the growth of the cells and the synthesis of metabolites, prolonging the fermentation cycle while reducing fermentation efficiency. Therefore, 0.2 g/L acetic acid feeding should be chosen for each intermittent feeding (Figure 4F). The production of 2,3-BD reached $21.83 \pm 0.56$ g/L, which increased the production of 2,3-BD by 61.46% and the yield of 2,3-BD by 58.51% compared to the unfed fermentation group (Table S5). In other reports, RSM was also used to optimize conditions such as dissolved oxygen and pH and combined with fed-batch fermentation to improve the 2,3-BD production with *K. pneumoniae*. The results show that the production of 2.3-BD was increased by 1.2 times at 70 h [40].

**Table 5.** Production of 2,3-BD under different acetic acid feeding concentrations.

| Additive Concentration (g/L) | 2,3-BD Production (g/L) | 2,3-BD Yield (g/g) | Single-Cell Production Intensity (g/g/g) |
|---|---|---|---|
| 0.1 | $16.85 \pm 0.57$ | $0.112 \pm 0.003$ | $2.060 \pm 0.008$ |
| 0.2 | $21.83 \pm 0.56$ | $0.149 \pm 0.004$ | $2.764 \pm 0.009$ |
| 0.3 | $19.17 \pm 0.49$ | $0.131 \pm 0.003$ | $2.492 \pm 0.012$ |
| 0.4 | $16.68 \pm 0.52$ | $0.112 \pm 0.004$ | $2.154 \pm 0.008$ |
| 0.5 | $16.62 \pm 0.52$ | $0.111 \pm 0.004$ | $2.146 \pm 0.012$ |

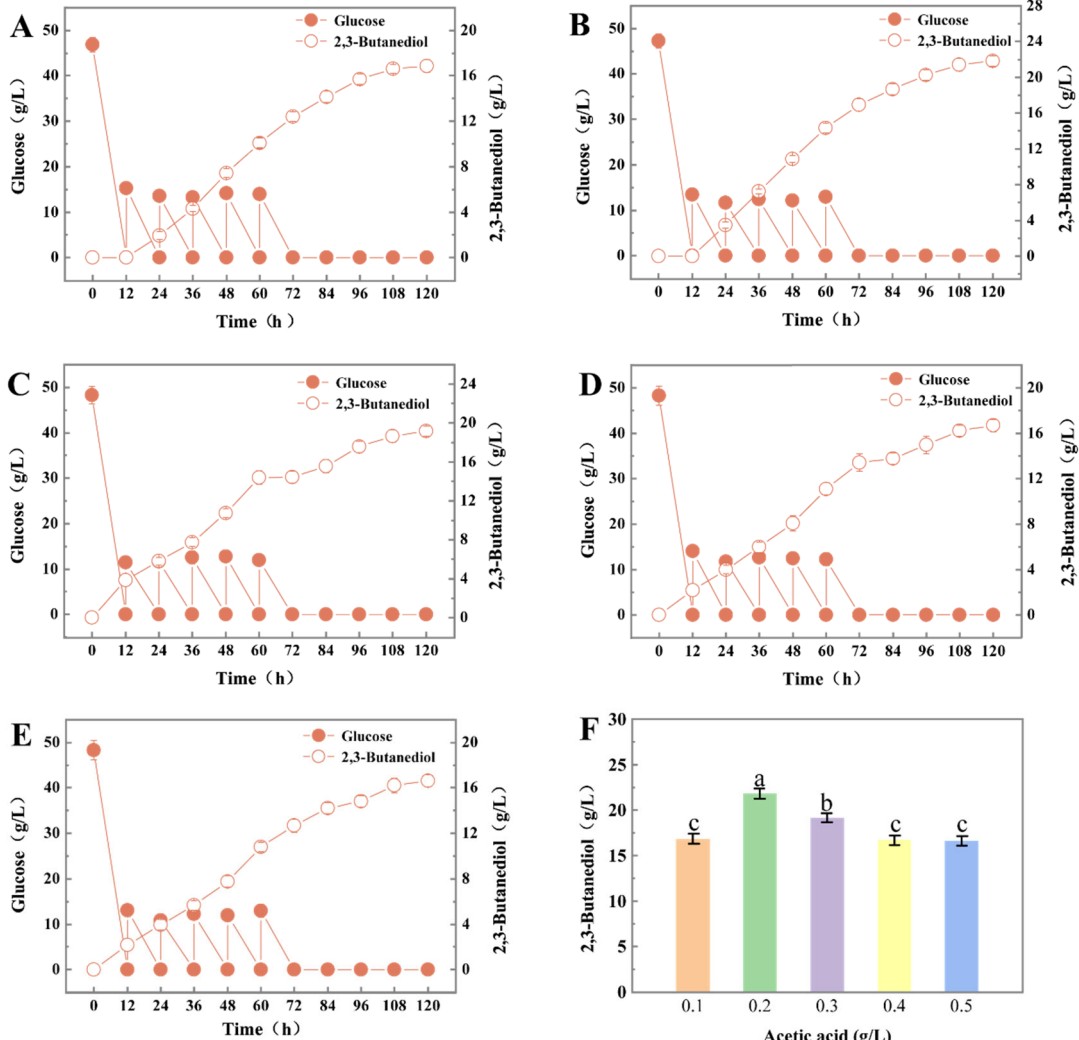

**Figure 4.** The effects of different feed amounts of acetic acid on the production of 2,3-BD by *S. cerevisiae* L7. (**A–E**) Acetic acid flow doses were 0.1, 0.2, 0.3, 0.4 and 0.5 g/L, respectively; (**F**) Comparison of the output of 2,3-BD with different acetic acid flow rates during 120 h fermentation (different letters represent significant differences between variables).

## 4. Conclusions

Three significant factors (acetic acid, peptone, glucose) affecting the production of 2,3-BD by *S. cerevisiae* L7 were determined with the PB design, steepest climbing experiment and CCD in this study. The most favorable medium composition for 2,3-BD production by *S. cerevisiae* L7 contained 145.58 g/L glucose, 21.58 g/L peptone and 0.72 g/L acetic acid and provided a maximal 2,3-BD production of 13.52 ± 0.12 g/L. Appropriately reducing the initial glucose content in the medium and feeding with glucose and acetic acid further increased the production of 2,3- BD to 21.83 ± 0.56 g/L and the yield to 0.15 ± 0.01 g/g. The outcomes of these studies indicate that *S. cerevisiae* L7 has the potential for the efficient production of 2,3-BD on a large scale.

**Supplementary Materials:** The following supporting information can be downloaded at: https://www.mdpi.com/article/10.3390/fermentation9070694/s1, Figure S1: A: The growth curve of *S. cerevisiae* L7 in fermentation medium. B: The dynamic changes of the 2,3-BD production and the total number of cells in the fermentation medium of *S. cerevisiae* L7. C: The changes of pH value of *S. cerevisiae* L7 in medium; Figure S2: Standard effect Pareto chart of factor to response value; Figure S3: Comparison of regression coefficients of influencing factors; Table S1: The two-level factorial design

in actual level and results; Table S2: The central composite design (CCD) in actual level and results; Table S3: Equality of Variances; Table S4: Result of T-test; Table S5: Experimental design and results of model validation.

**Author Contributions:** G.A. and S.S.: Methodology, Data curation, Writing—original draft, Writing—review and editing; L.L.: Formal analysis; Y.G.: Investigation; X.T.: Validation; W.P.: Resources, Supervision; J.G.: Resources, Supervision and Funding. All authors have read and agreed to the published version of the manuscript.

**Funding:** This research was funded by the National Natural Science Foundation of China (No. 32071519); the Key Program of Heilongjiang Provincial Natural Science Foundation of China (No. ZD2020C008); Scientific Research Project of Ecological Environment Protection of Heilongjiang Provincial Department of Ecological Environment (HST2022TR004); Open project of Hebei Key Laboratory of Agroecological Safety (2023SYSJJ18).

**Institutional Review Board Statement:** Not applicable.

**Informed Consent Statement:** Not applicable.

**Data Availability Statement:** The data presented in this study are available upon request from the corresponding author.

**Conflicts of Interest:** The authors declare no conflict of interest.

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
