# Peer review of "Production of 2,3-Butanediol by S. cerevisiae L7 in Fed-Batch Fermentation with Optimized Culture Conditions"

_fermentation, doi:10.3390/fermentation9070694_

Round 1

Reviewer 1 Report

The manuscript focuses to optimize the biotechnological production of 2,3-butanediol, a bulk chemical with a wide range of industrial applications. The production of this diol by fermentation is in accordance with the new concept of “white biotechnology” aiming to produce chemicals from renewable resources. The subject is very interesting and very useful. A wild type Saccharomyces cerevisiae was used and the optimization involved the selection of the best culture medium composition and culture conditions, through the use of experimental design tools to achieve the best conditions to maximize the production rate and yield of the target compound.

It would have been very interesting to confirm the medium and operational conditions that drove the best production of 2,3-BD with working volume of 100 mL, in a larger scale bioreactor with mechanical agitation, to be closer to the industrial scale conditions of operation.

The manuscript has a high potential to be published but needs a strong revision to improve the structure and style of English language, as well as a great effort to clarify the justification and the interpretation of results. Some parts of the text are confused and difficult to interpret.

Examples:

-          Line 16 and 17 - …. were obtained ….

-          Line 41 - … the lower production …

-          Line 114 – And the values ….

-          Line 170 - … would not be considered in subsequent.

-          Line 170 - … was significantly affected the production …

-          Line 190 – … were used to selected the …

-          Line 183 - … with solid stopper to regulate optimal oxygen …

-          Line 219 - … were in Table 2.

-          Line 270 - … interaction between peptone and glucose …

-          Line 306 - … production were rough similarity with ….

Line 21 – yield efficiency - 0.15 g/g – g of 2,3-BD per g (of what?). Also in Table 5 explain the meaning of yield.

Line 23 – this finding supply good expectations for large-scale ….; the same in Line 84

Line 41 – Why the mention of refª 12, from the own authors, when they are speaking on the use of yeast and the refª 12 refers to the use of a bacterium?

Line 46 – a yield is not expressed in g/L! ; the same in Line 53.

Line 52 – use “mL “ in accordance to the symbol “L” for litre.

Line 53 – what is the meaning of “delta l dh A “ after the name of Klebsiella pneumoniae ?

Line 55 – g/g is a yield, and not a production.

Line 73 – did you really add “acetoin” to your culture medium?

Line 76 - … other components – must be specified which are they.

Line 78 - … “ingenious design” – is probably not the best designation!

Line 81 - …. The purpose of this study was to …

Line 106 – why PB? The same in Line 114 - … “by PB (?)”; the same in Line 115 (say the meaning of PB); also Lines 121, 190, 211, 359

Table 1 – What was the rationale to choose the range values of each factor? It is not referred in the text.

Line 135 – “carbon source” means glucose? Or the sum of glucose + peptone?

Line 137 - … flasks with working volume of … (a “w” is missing”)

Line 139 - … the appropriate time of acetic acid addition (and glucose?)

Line 146 – Is it necessary the refª 30 (from the authors) here to support a flow rate of the mobile phase (a very common value…)?

Line 177 - … at two-stage shaking speed …. – explain the meaning of this, please. It is explained in 2.3 of supplementary file but it should be included in the main text.

In Figure 1, when checking the effect of each single factor on 2,3-BD production, what were the values of the other components kept constant? It is explained in 2.3 of supplementary file but it should be included in the main text.

Line 191, 192 - … the values of 2,3-BD production for the fifteen runs were presented in Table 1 (?)

Lines 210 – 222 – explain better, please.

Line 228 – these values of production are not represented in the main text

Line 237 - Table 3 – in the header title, do you really mean “atrazine” (that is an herbicide)?

Line 276 – Yu et al – add the number of the refª, please

Line 284 – Narisetty et al - add the number of the refª, please

Lines 288 – 295 – explain better, please

Line 296 – title of item 3.5 - … in rotary shake-flask

Line 299 – (Fig 3) does not support what is written in line 299!

Line 302 + Lines 323 – 325 – we cannot clear understand the feeding sequence; I think that the times reported in Line 302 refer to the first addition of glucose + acetic acid, because the interval after that, for other additions, was always 12 h. Re-written the legend of Figure 3, please.

Line 307 – Fig 3 does not show the assay without fed-batch feeding!

Lines 308 – 312 – explain better, please

Line 317 – when glucose and acetic acid were feeding at 12 h (for the first time?)

Graphs of Figure 4 – the symbols are not correct – the glucose is the full ball and the 2,3-BD is the empty ball!

Line 324 - Figure 3 – in the legend - … A, B, C and D were (?) glucose ….

Line 325 – Figure 3 - … under different acetic acid feed timings (?)

Table 4 and Table 5 – what is the concept of “Single cell production intensity” and the associate unity of “g/g/g” ?!

Line 336 – Liu et al. …. The number of the reference is missing.

Lines 328 – 337 – explain better; in Fig 4 is there no addition of glucose?

Lines 338 – 350 – explain better and support with references.

In Fig 4 the feeding strategy was different from the one in Fig 3. In Fig 4 the feeding time is constant and the acetic acid dosage is increasing. This is not evident when reading the text.

Line 347 - ….unfeeding fermentation group. In other reports, RSM also used to optimize … … (?)

Line 364 – meaning of yield?

Line 365 – had the potential (or has?)

Additional notes:

-          Through the graph B of Fig S1 A we can observe that the 2,3-BD is a secondary metabolite; its synthesis only starts when cell growth profile achieves the stationary phase.

-          In the graph C of Fig S1 A we can observe that the pH of the culture decreases along the fermentation – why? The acetic acid is supposed to be consumed?!

A strong revision to improve the structure and style of English language is needed.
